# Differential voltage-dependent modulation of the ACh-gated K$^+$ current by adenosine and acetylcholine

Ana Laura López-Serrano[1], Rodrigo Zamora-Cárdenas[1], Iván A. Aréchiga-Figueroa[2], Pedro D. Salazar-Fajardo[1], Tania Ferrer[1], Javier Alamilla[3], José A. Sánchez-Chapula[1], Ricardo A. Navarro-Polanco[1], Eloy G. Moreno-Galindo[1]*

1 Centro Universitario de Investigaciones Biomédicas, Universidad de Colima, Colima, Col., Mexico,
2 CONACYT, Facultad de Medicina, Universidad Autónoma de San Luis Potosí, San Luis Potosí, S.L.P., Mexico, 3 CONACYT, Centro Universitario de Investigaciones Biomédicas, Universidad de Colima, Colima, Col., Mexico

☯ These authors contributed equally to this work.
* eloy@ucol.mx

**Data Availability Statement:** All relevant data are within the paper and its supporting information files.

## Abstract

Inhibitory regulation of the heart is determined by both cholinergic M$_2$ receptors (M$_2$R) and adenosine A$_1$ receptors (A$_1$R) that activate the same signaling pathway, the ACh-gated inward rectifier K$^+$ (K$_{ACh}$) channels via G$_{i/o}$ proteins. Previously, we have shown that the agonist-specific voltage sensitivity of M$_2$R underlies several voltage-dependent features of $I_{KACh}$, including the 'relaxation' property, which is characterized by a gradual increase or decrease of the current when cardiomyocytes are stepped to hyperpolarized or depolarized voltages, respectively. However, it is unknown whether membrane potential also affects A$_1$R and how this could impact $I_{KACh}$. Upon recording whole-cell currents of guinea-pig cardiomyocytes, we found that stimulation of the A$_1$R-G$_{i/o}$-$I_{KACh}$ pathway with adenosine only caused a very slight voltage dependence in concentration-response relationships (~1.2-fold EC$_{50}$ increase with depolarization) that was not manifested in the relative affinity, as estimated by the current deactivation kinetics ($\tau$ = 4074 ± 214 ms at -100 mV and $\tau$ = 4331 ± 341 ms at +30 mV; $P$ = 0.31). Moreover, $I_{KACh}$ did not exhibit relaxation. Contrarily, activation of the M$_2$R-G$_{i/o}$-$I_{KACh}$ pathway with acetylcholine induced the typical relaxation of the current, which correlated with the clear voltage-dependent effect observed in the concentration-response curves (~2.8-fold EC$_{50}$ increase with depolarization) and in the $I_{KACh}$ deactivation kinetics ($\tau$ = 1762 ± 119 ms at -100 mV and $\tau$ = 1503 ± 160 ms at +30 mV; $P$ = 0.01). Our findings further substantiate the hypothesis of the agonist-specific voltage dependence of GPCRs and that the $I_{KACh}$ relaxation is consequence of this property.

## Introduction

Stimulation of both cholinergic muscarinic M$_2$ receptors (M$_2$R) and adenosine A$_1$ receptors (A$_1$R) has an important physiological impact on the electrophysiology and mechanical

**Funding:** This work was supported by SEP-CONACYT, Mexico. Grant No. CB-2011-01-167109 (to E.G.M-G.), and CB-2013-01-220742 (to R.A.N-P). The funders had no role in study design, data collection and analysis, decision to publish, or preparation of the manuscript.

**Competing interests:** The authors have declared that no competing interests exist.

function of the heart by acting on the same downstream signaling pathway. These receptors activate, via pertussis-toxin-sensitive G-proteins ($G_{i/o}$), the acetylcholine (ACh)-gated inward rectifier $K^+$ ($K_{ACh}$) channels [1–4], which are composed of the G-protein-coupled inwardly rectifying $K^+$ channel subunits, Kir3.1 and Kir3.4 [5]. Once stimulated by $M_2R$ or $A_1R$, $G_{i/o}$ proteins release $G\alpha$ subunits to ultimately inhibit the cardiac contractility by a cAMP-dependent mechanism, whereas $G\beta\gamma$ subunits directly activate $K_{ACh}$ channels to produce electrophysiological effects, such as slowing the heart rate, reducing the action potential duration and the effective refractory period, hyperpolarizing the resting membrane potential, and prolonging the spontaneous diastolic depolarization [2, 4, 6].

Recently, in cat atrial myocytes we have shown that $M_2R$ exhibits agonist-specific voltage dependence, where the intrinsic voltage sensitivity of this receptor [7–9] modifies its affinity for diverse agonists in a ligand-selective manner, which is eventually reflected on the activation of the coupled $K_{ACh}$ channels [10]. This property can be distinguished in the deactivation kinetics of the current carried by these channels, $I_{KACh}$ [11]. Also, we previously proposed that this is the molecular mechanism underlying a very distinctive attribute of receptor-stimulated Kir3.x currents (including $I_{KACh}$), known as *relaxation* [12], which consists of a time-dependent augment or reduction of the current upon hyperpolarizing or depolarizing, respectively, the cardiomyocyte membrane with voltage steps [13, 14]. Agonists with higher affinity at hyperpolarized potentials induce $I_{KACh}$ to manifest its typical relaxation behavior, whereas those ligands with inverse voltage dependence, i.e., higher affinity at depolarized voltages, lead to this current to display the appearance of an "opposite" relaxation, with delayed rectifying characteristics [12, 15, 16].

Alike as for $M_2R$, voltage sensitivity is an emerging property for several other G-protein-coupled receptors (GPCRs) [17–24]. In cardiomyocytes, $I_{KACh}$ is the effector for $M_2R$ but also for $A_1R$ and it is currently unknown how voltage affects this latter and how this important potassium current is modulated in consequence. However, in cat atrial cells the signaling pathway $A_1R$-$G_{i/o}$-$I_{KACh}$ was not functional in our conditions. Therefore, in this work we used guinea-pig atrial myocytes (where this pathway is operative [25, 26]) to assess the effects of membrane potential on the $I_{KACh}$ activation and the ability to induce a voltage-dependent hallmark of the current (relaxation) when the signaling pathway is triggered by adenosine (Ado) through $A_1R$.

## Materials and methods

### Ethics

Animals were treated humanely following the Guide for the Care and Use of Laboratory Animals published by the US National Institutes of Health (NIH Publication No. 85–23, revised 1996). The experimental protocol was approved by the Institutional Animal Care Committee of the University of Colima.

### Isolation of atrial myocytes

Single myocytes were obtained from the left atrium of adult guinea pigs of either sex (400–600 g) by collagenase/protease enzymatic perfusion as previously described [25]. Guinea pigs were anaesthetized with sodium pentobarbitone (40 mg/kg, I.P.), heparinized (1000 U/kg, I.P.), and then euthanized by excision of the heart *en bloc* when deeply anesthetized. The lack of pedal withdrawal reflexes was used to check the extent of anesthesia. Isolated myocytes were kept in Kraft-Brühe solution at 4°C for 2–12 h before being used for electrophysiological recordings. The composition of the Kraft-Brühe solution was (in mM): 80 K-glutamate, 40 KCl, 20 taurine,

10 $KH_2PO_4$, 5 $MgSO_4$, 10 glucose, 10 Hepes, 0.5 creatine, 10 succinic acid, and 0.2 EGTA; pH was adjusted to 7.4 with KOH. This solution was bubbled with 100% $O_2$.

## Electrophysiological recordings

$I_{KACh}$ was recorded from atrial myocytes by means of the whole-cell configuration of the patch-clamp technique. Recordings were obtained using an Axopatch-200B amplifier and a Digidata 1440 A digitizer (Molecular Devices, Sunnyvale, CA, USA), while pulse generation and data acquisition were done using the pCLAMP 10 software (Molecular Devices). Patch pipettes were prepared from borosilicate capillary glass (WPI, Sarasota, FL, USA) and had tip resistance between 1.5 and 3 MΩ. The pipette solution contained (mM): 80 K-aspartate, 20 KCl, 10 $KH_2PO_4$, 5 Hepes, 5 $K_4BAPTA$, 1 $MgSO_4$, 0.2 Na-GTP, and 3 $Na_2ATP$; pH was adjusted to 7.25 with KOH. Currents were filtered with a four-pole Bessel filter at 1 kHz and digitized at 5 kHz. The bath was grounded through an agar-KCl bridge. Capacitance and series resistance were compensated to minimize the duration of the capacitive current.

For recordings, the extracellular solution contained (in mM): 136 NaCl, 4 KCl, 10 Hepes, 0.5 $CoCl_2$, 1 $MgCl_2$, 0.1 $CaCl_2$, and 11 glucose (pH was adjusted to 7.35 with NaOH). The rapid delayed rectifier ($I_{Kr}$) and slow delayed rectifier currents ($I_{Ks}$) were blocked by 3 μM E-4031 and 50 μM chromanol 293B, respectively. Recordings for concentration-response (C-R) relationships and for the estimation of the $I_{KACh}$ deactivation kinetics (tau) were carried out at room temperature (22–24˚C) and according to previous publications [9, 11]. For these two approaches, inward rectifier current ($I_{K1}$) was greatly reduced by adding 2 μM $BaCl_2$ in the recording extracellular solution [27]. In addition, a fast perfusion system composed of a triple-barrel glass pipette controlled by an electromechanical switching device (SF-77B, Warner Instruments, Hamden, CT, USA) was used to exchange bath (extracellular) solutions within ~ 250 ms. Voltage-step (square-pulse) protocols were performed at 36 ± 0.5˚C and using a standard perfusion system (exchange rate ~1 mL/min) to apply the external solution containing Ado or ACh until reaching steady-state effects, and thereby $I_{KACh}$ was obtained by digitally subtracting control currents from those obtained in the presence of these agonists.

## Drugs

Ado and ACh (Sigma-Aldrich, St Louis, MO, USA) were dissolved in deionized water to make 10 mM stock solutions and stored at -20˚C. Working agonist concentrations were freshly prepared in the recording extracellular solution. E-4031 and chromanol 293B were obtained from Tocris Bioscience (Ellisville, MO, USA), $K_4BAPTA$ was acquired from Santa Cruz Technology (Dallas, TX, USA), and all other reagents were from Sigma-Aldrich.

## Data analysis

Data analysis was done using pCLAMP 10 (Molecular Devices) and Origin 8 software (OriginLab Corp., Northampton, MA, USA). $I_{KACh}$ deactivation time constants were obtained by fitting a single exponential equation to the current traces. For C-R relationships, the normalized amplitude of $I_{KACh}$ (E) was plotted as a function of the Ado or ACh concentration ([X]). These data were fitted to a Hill equation: $E = \frac{Emax*X^{nH}}{EC_{50}{}^{n}+X^{nH}}$ to estimate $EC_{50}$ (the half-maximal effective concentration), nH (Hill coefficient), and $E_{max}$ (the maximum asymptotic value). $pEC_{50}$ was calculated as the negative logarithm of the $EC_{50}$.

## Statistical analysis

Results are reported as mean ± SEM ($n$ = number of cardiomyocytes). Statistical analyses were performed using the Origin 8 software (OriginLab Corp.). Statistical difference was evaluated by the paired $t$ test after verifying the normal distribution of data with the Shapiro-Wilk test. Otherwise, the Wilcoxon signed rank test was applied. A $P$ value less than 0.05 was considered as significant.

## Results

### Null or very weak voltage-dependent effects of Ado on the cardiac $A_1R$-$G_{i/o}$-$I_{KACh}$ signaling pathway

In previous studies, we described the agonist-specific voltage sensitivity of $M_2R$, where membrane depolarization reduces, augments, or not modifies the receptor activation by several muscarinic agonists [9, 11, 16]. Here, we investigated whether $A_1R$ exhibits a voltage-dependent interaction with its physiological agonist Ado, by measuring $I_{KACh}$ activation in guinea pig cardiomyocytes. The effects of different Ado concentrations on $I_{KACh}$ at +30 and -100 mV are shown in Fig 1A. The C-R relationship was marginally shifted to hyperpolarized potentials, displaying a slight but statistically significant ($P$ = 0.03; paired $t$ test) greater potency at the negative potential (Fig 1B and S1 Table). The pEC$_{50}$ for current activation at +30 mV was 6.54 ± 0.08 (288 nM), while at -100 mV was 6.62 ± 0.07 (240 nM), thereby a ~1.2-fold EC$_{50}$ increase with depolarization.

Next, we analyzed the effect of voltage on the deactivation kinetics of $I_{KACh}$, as this reflects the rate of agonist unbinding, and is therefore a comparative index of affinity [28–30]. Fig 2A depicts current traces of $I_{KACh}$ elicited by 10 µM Ado recorded at +30 and -100 mV. The time

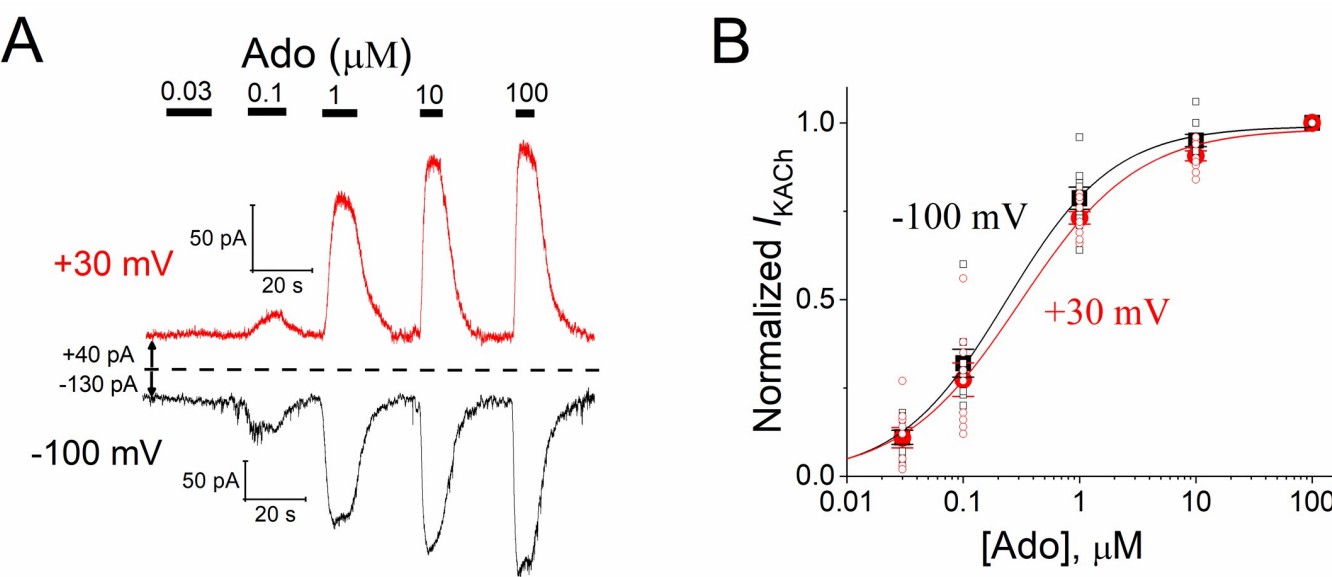

**Fig 1. Effects of membrane potential on $I_{KACh}$ evoked by Ado through $A_1R$ in guinea pig atrial myocytes. a)** Examples of $I_{KACh}$ traces elicited when sequentially perfusing the cells with increasing Ado concentrations and recorded at $V_h$ of +30 mV (red traces) and -100 mV (black traces). The zero current level is indicated by the dashed line. **b)** C-R relationships for $I_{KACh}$ activated by Ado at $V_h$ +30 mV (red circles) and -100 mV (black squares). The filled symbols represent the current amplitudes normalized to that activated by 100 µM Ado. The solid lines depict the best data fit to a Hill equation. The pEC$_{50}$ values are reported in the text. The maximum asymptote (0.98 ± 0.01 at +30 mV and 1.00 ± 0.01 at -100 mV) was not significantly altered by voltage (P = 0.16; Wilcoxon signed rank test) neither was the Hill coefficient (1.03 ± 0.12 at +30 mV and 1.02 ± 0.09 at -100 mV) (P = 0.98; paired $t$ test). $n$ = 9 myocytes. For this and subsequent figures, the filled symbols represent the mean values, while the open symbols denote the individual raw data points.

course of current deactivation was not significantly different at the two holding potentials evaluated (Fig 2B): 4331 ± 341 ms (+30 mV) and 4074 ± 214 ms (at -100 mV; $P = 0.31$; paired $t$ test), suggesting that the affinity agonist-receptor, assayed by this approach, is not affected by the membrane voltage.

Afterwards, we assessed whether the classical $I_{KACh}$ relaxation can be induced by the activation of $A_1R$ with Ado, as is characteristic when sub-saturating concentrations of agonist are used [12, 13, 17, 31]. For this purpose, the currents evoked by a sub-saturating (0.3 μM) and a saturating (30 μM) Ado concentration were recorded at -110 mV for 2.5 s, after pre-pulses of 2.5 s between -110 and +50 mV, with 20-mV increments, from a holding potential ($V_h$) of -50 mV (Fig 3A and 3B). Interestingly, in our experimental conditions the typical voltage-dependent relaxation of $I_{KACh}$ was not generated by any of Ado concentrations, and thus currents at -110 mV (and even those obtained with the pre-pulses) did not exhibit time-dependence, that is to say, a gradual change in current amplitudes was not perceived. This effect was quantified in Fig 3C, where it is illustrated the relationship between the instantaneous current ($I_{ins}$) at the beginning of the pulse at -110 mV in respect to the maximal current ($I_{max}$) at the end of the pulse, as a function of the pre-pulse potential. The fraction of open channels at the pre-pulse potentials, denoted by the ratio $I_{ins}/I_{max}$, remained virtually unchanged and it was very similar for both Ado concentrations.

## Stimulation of the $M_2R$-$G_{i/o}$-$I_{KACh}$ signaling pathway by ACh is voltage-dependent

Given the null or very weak influence of voltage on the $I_{KACh}$ induced by Ado through $A_1R$ in guinea-pig cardiomyocytes, we also assessed the effects of ACh on the $I_{KACh}$ activation through $M_2R$ in this species using the same experimental conditions (Fig 4A). With ACh-$M_2R$, the C-R relationship obtained at -100 mV was also shifted to the left in comparison to that at +30 mV. However, in contrast to the very weak (EC$_{50}$ change ~1.2-fold) influence of voltage with Ado-$A_1R$, the voltage-dependent effect to activate $I_{KACh}$ was stronger with ACh-$M_2R$, since at -100 mV pEC$_{50}$ = 6.34 ± 0.10 (457 nM), whereas at +30 mV pEC$_{50}$ = 5.90 ± 0.06 (1259 nM)

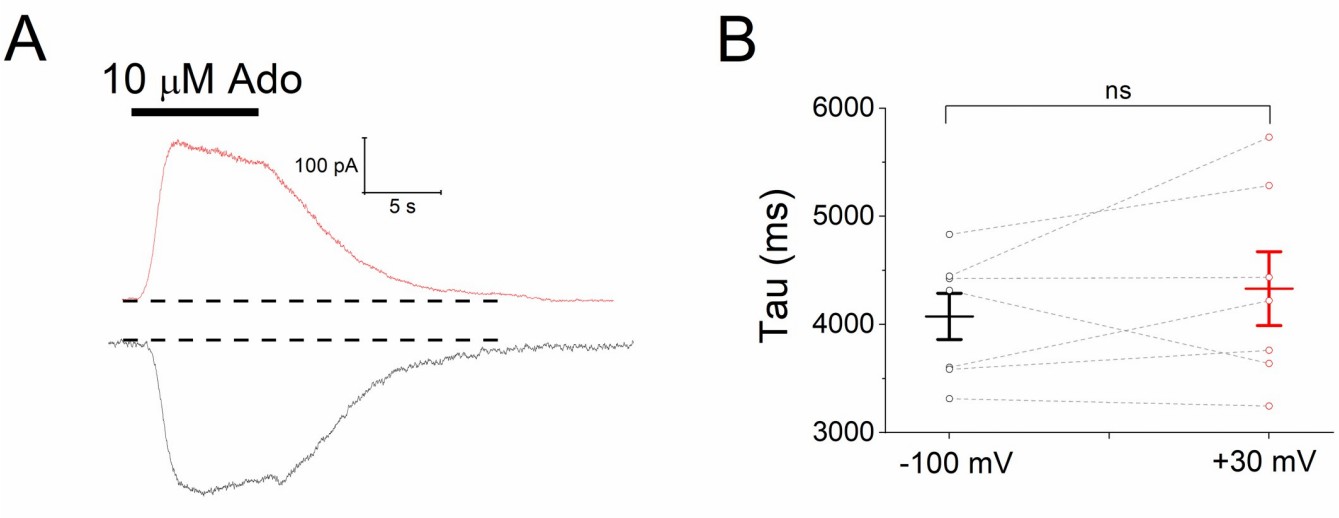

**Fig 2. Deactivation of $I_{KACh}$ induced by Ado is not voltage-dependent. a**) Current recordings ($I_{KACh}$) activated by 10 μM Ado at +30 (red trace) and -100 mV (black trace) to estimate the deactivation kinetics (**b**). For **a**, the dashed lines designate the basal current level. $n$ = 7 myocytes. ns = not significant.

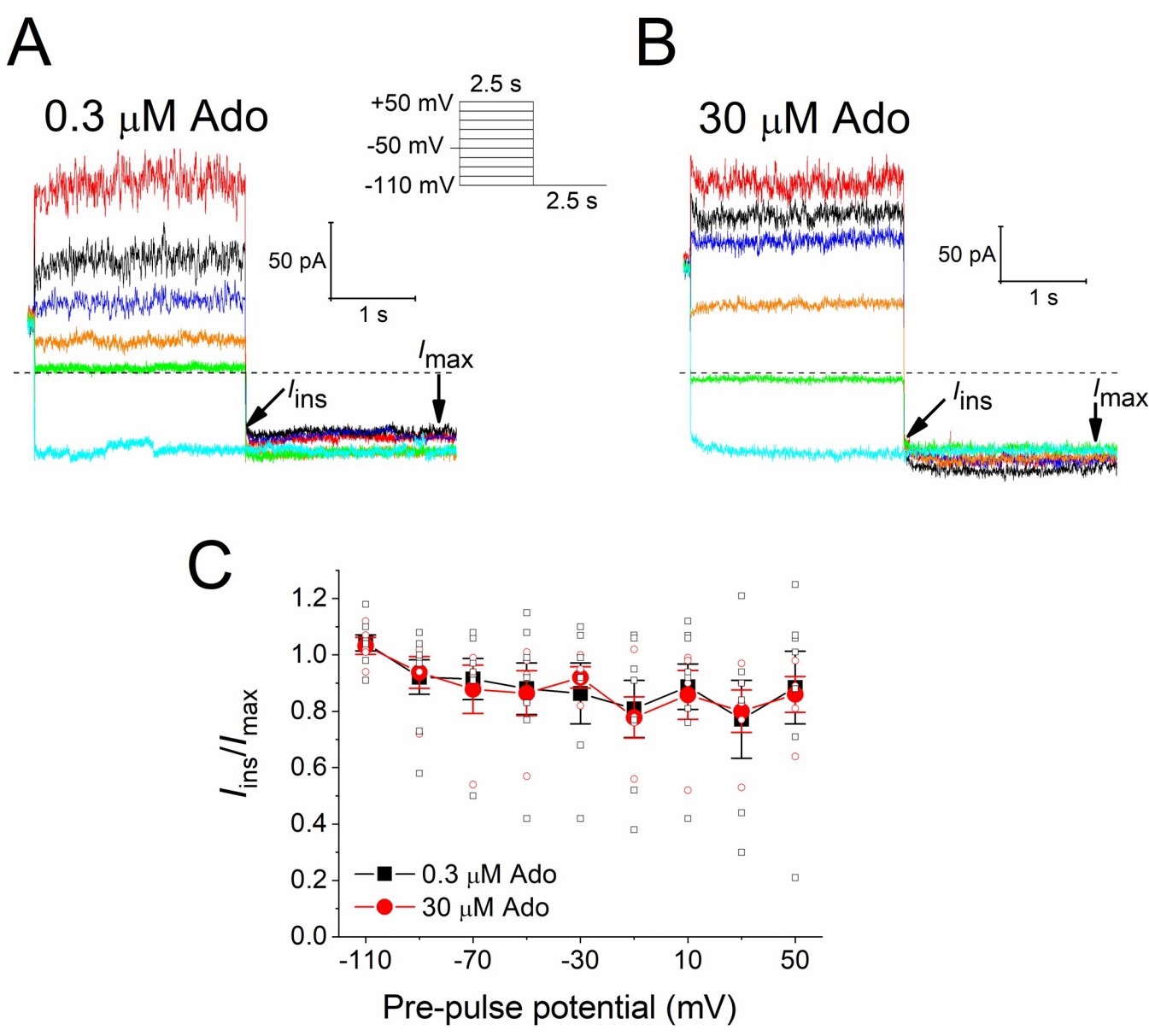

**Fig 3. $I_{KACh}$ relaxation is not induced by Ado.** Illustrative steady-state current traces evoked by 0.3 (**a**) and 30 μM (**b**) Ado using the square wave voltage protocol shown in the inset. For clarity, currents at voltages -110, -90, -70, -30, +10, and +50 mV are only shown. $I_{ins}$ and $I_{max}$ are explained in the text. **c)** $I_{ins}/I_{max}$ ratio against the pre-pulse potential for the currents elicited by 0.3 ($n = 8$) and 30 μM Ado ($n = 5$).

($P = 0.01$; paired $t$ test), and hence a ~2.8-fold $EC_{50}$ increase with depolarization (Fig 4B and S2 Table).

Contrary to the results obtained with Ado, we observed that membrane potential affected the deactivation kinetics of $I_{KACh}$ when evoked by ACh-$M_2$R. Indeed, the time course of $I_{KACh}$ deactivation with ACh was significantly slower at -100 mV compared to that at +30 mV (Fig 5A). Time constants for current deactivation were 1762 ± 119 ms at -100 mV *versus* 1503 ± 160 ms at +30 mV ($P = 0.01$; paired $t$ test) (Fig 5B), suggesting a higher affinity of ACh for $M_2$R at hyperpolarized potentials.

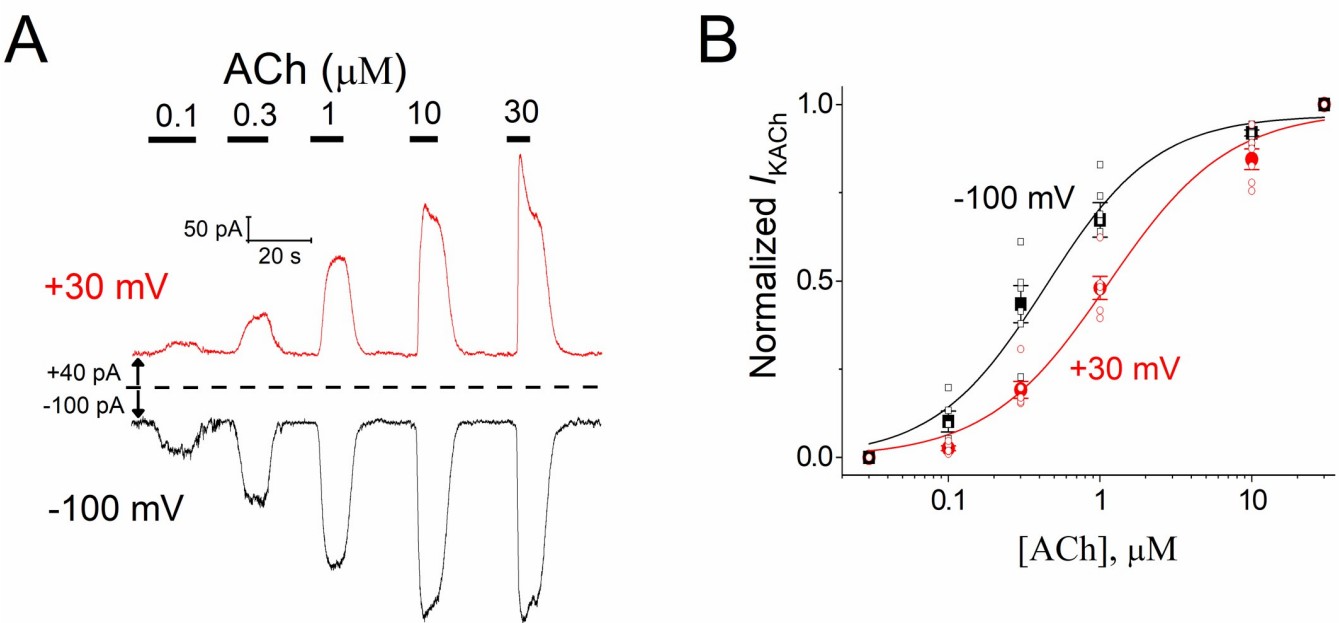

**Fig 4. Activation of $I_{KACh}$ by ACh is voltage-dependent. a**) Representative $I_{KACh}$ recordings evoked at +30 mV (red traces) and -100 mV (black traces) by increasing concentrations of ACh. The dashed line symbolizes the zero current level. **b**) C-R curves for $I_{KACh}$ activation by ACh at both voltages studied (red squares at +30 mV and black circles at -100 mV). Data were fitted to a Hill equation (solid lines). pEC$_{50}$ are mentioned in the text. Neither the maximum asymptote (1.01 ± 0.01 at +30 mV and 0.98 ± 0.01 at -100 mV) nor the Hill coefficient (1.04 ± 0.06 at +30 mV and 1.23 ± 0.07 at -100 mV) were significantly modified by voltage ($P$ = 0.14 and 0.13, respectively; paired $t$ test). $n$ = 6 myocytes.

Finally, to verify if the $I_{KACh}$ relaxation property of $I_{KACh}$ can be generated by ACh-M$_2$R, we used the same double-pulse voltage protocol as that for Ado-A$_1$R (inset of Fig 6). With the sub-saturating ACh concentration (0.3 μM), outward currents obtained at depolarized pre-pulses exhibited an instantaneous component followed by a slow time-dependent decrease, whereas hyperpolarization to -110 mV induced a gradual increase of inward currents (Fig 6A).

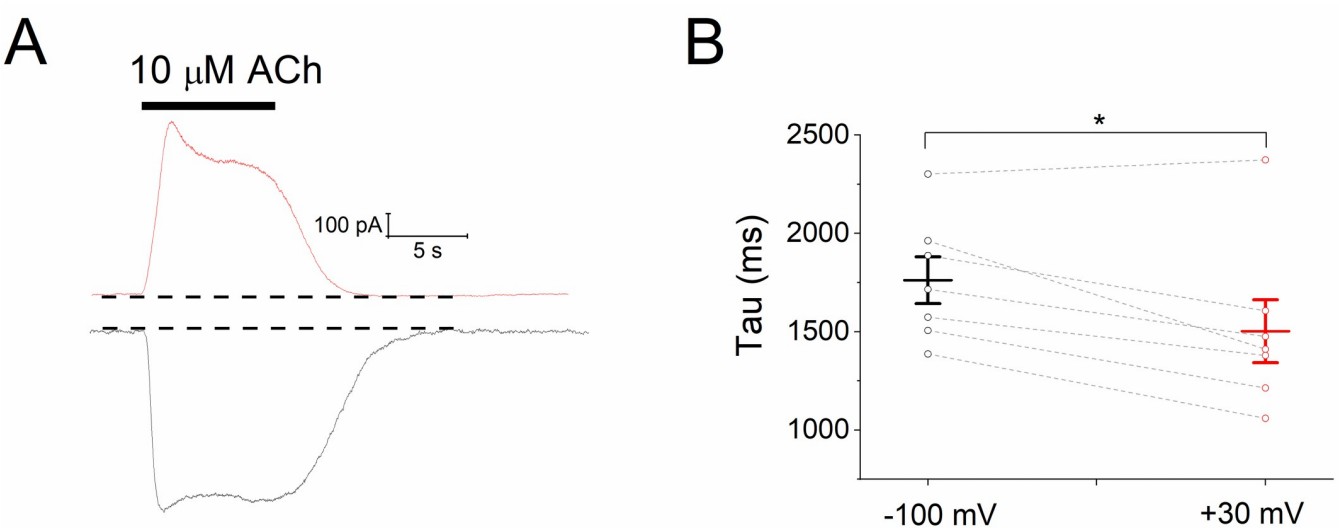

**Fig 5. Voltage-dependent deactivation of $I_{KACh}$ when it is activated by ACh. a**) $I_{KACh}$ generated by 10 μM ACh at +30 (red trace) and -100 mV (black trace) to measure the time course of the deactivation process (**b**). The basal current level is indicated by the dashed lines in panel **a**. $n$ = 7 myocytes. $^*$, $P$ = 0.01.

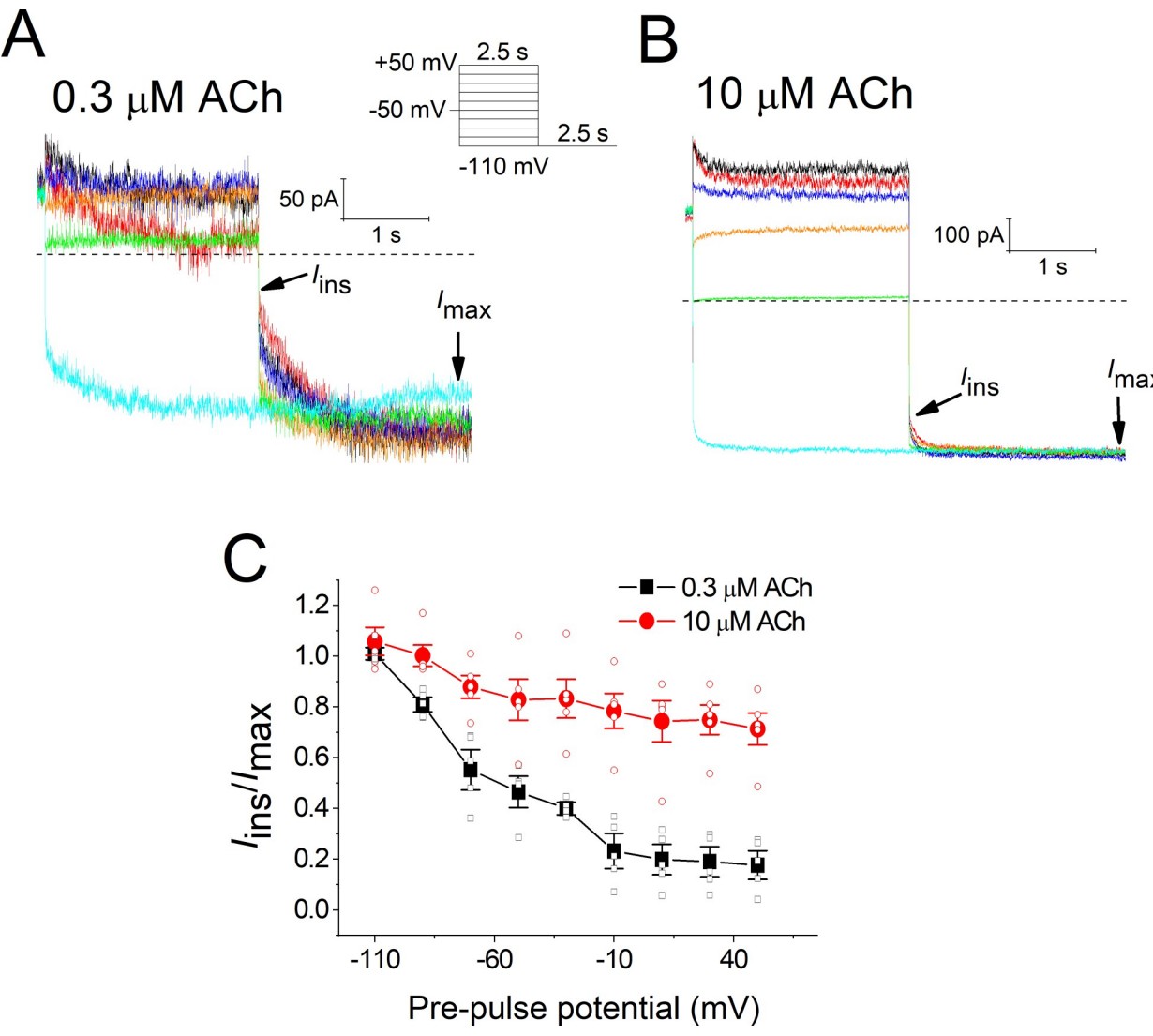

**Fig 6.** $I_{KACh}$ **relaxation is induced by ACh-M$_2$R. a**) Typical steady-state $I_{KACh}$ traces obtained in the presence of 0.3 (A) and 10 μM ACh (**b**) with a voltage protocol (inset). For clarity, current traces evoked at voltages -110, -90, -70, -30, +10, and +50 mV are only depicted. **c**) Fraction of open channels (I$_{ins}$/I$_{max}$) as a function of the pre-pulse potential for the currents evoked by 0.3 and 10 μM ACh. $n$ = 5 for each ACh concentration.

This typical behavior, the $I_{KACh}$ relaxation, was strikingly reduced in the presence of 10 μM ACh, a nearly saturating concentration of this agonist (Fig 6B). With 0.3 μM ACh, the fraction of open channels (I$_{ins}$/I$_{max}$) progressively diminished as pre-pulse potentials were more depolarized, but this effect was greatly attenuated with 10 μM ACh (Fig 6C). These data clearly show that, in contrast to Ado, ACh is able to induce the relaxation property of $I_{KACh}$.

## Discussion

Despite M$_2$R and A$_1$R activate the same downstream signaling pathway in cardiac myocytes, in this work we found a differential voltage-dependent modulation of $I_{KACh}$ by the physiological agonists ACh and Ado, that is, a null or very weak influence of voltage on the $I_{KACh}$ evoked by Ado-A$_1$R that made this current to display no relaxation behavior. By contrast, with ACh-

M$_2$R a clear voltage-dependent effect was observed on $I_{KACh}$, as well as the development of the characteristic relaxation of the current.

Our results herein support the agonist-specific voltage sensitivity of GPCRs [9, 11, 21, 22, 32, 33]. Indeed, the slower $I_{KACh}$ deactivation kinetics at hyperpolarized potentials (Fig 5) indicates a greater affinity of ACh towards M$_2$R [8, 28, 30] that explains the leftward shift of the C-R relationship (Fig 4). This correlates with the time-dependent increase of the current upon hyperpolarization; and the opposite with depolarization, thereby giving rise to the relaxation process of $I_{KACh}$ (Fig 6A and 6C). This property is strikingly reduced by high ACh concentrations (Fig 6B and 6C) due to the maximal activation of K$_{ACh}$ channels both at negative and positive potentials. These results are consistent with those obtained in feline cardiomyocytes with the superagonist iperoxo [34]. In the activation of the A$_1$R-G$_{i/o}$-$I_{KACh}$ pathway with Ado, the slight voltage dependence detected in the C-R curves (Fig 1) was not reflected in the relative affinity assessed by the current deactivation kinetics (Fig 2) [29, 30]; and thus time-dependent changes of the current upon modifying the membrane voltage (relaxation) were not observed (Fig 3).

As a molecular mechanism, our findings are not consistent with the concept that M$_2$R voltage dependence arises from voltage-induced transitions between the high-and low-affinity states of the receptor when coupled or not to its cognate G-protein, respectively [8, 28, 35, 36], since this view is contradictory with evidence showing diverse voltage-dependent effects of different agonists on the same GPCR [11, 15, 16, 21, 22, 32, 33]. Alternatively, our data are more compatible with the idea that membrane potential induces conformational changes directly at the agonist binding (orthosteric) site of GPCRs, independent of G-protein coupling, which determine the modulation of the remainder signaling pathway [9, 11, 15, 20, 22]. Thus, regardless of the same effector of both cardiac signaling pathways, in M$_2$R the hyperpolarization-induced conformational changes (at the orthosteric site and the external access [34, 37]) provoke an increase in the affinity for ACh [8, 11, 28] that yields a higher stimulation of downstream cellular elements. In A$_1$R, it is conceivable that voltage also alters its molecular conformation but in such a manner that only a very slight change in the potency (probably the affinity) for Ado is generated (Fig 1), which is not mirrored in other less sensitive functional assays (deactivation kinetics and currents evoked with voltage step protocols). Interestingly, although Ado activates A$_1$R-$I_{KACh}$ with very slight voltage dependence, our results implicate that A$_1$R possesses voltage sensitivity, which suggests that other adenosine agonists could induce disparate voltage-dependent patterns, as evidenced by the agonist-specific voltage sensitivity of other GPCRs. Using here guinea-pig cardiomyocytes as a cellular model, where A$_1$R and M$_2$R converge in the same effector ($I_{KACh}$), makes more evident how voltage sensitivity of GPCRs is determinant in the remote modulation of downstream cellular effectors [38].

RGS proteins, particularly RGS4, has been considered responsible for the $I_{KACh}$ relaxation because this property only emerges if this protein is expressed in *Xenopus* oocytes reconstituted with the other main components of the muscarinic $I_{KACh}$ pathway, i.e., Kir3.1/Kir3.4 subunits and M$_2$R [13, 39–41]. Nevertheless, it has been previously shown that relaxation can still arise in oocytes (expressing Kir3.1/Kir3.4 and M$_2$R) that lack RGS4, although on a slower timescale [31], which indeed reinforces the critical role of RGS proteins in the regulation of $I_{KACh}$ kinetics, but convincingly demonstrating that they are not the determining factor of the relaxation mechanism. Furthermore, changes in the intracellular Ca$^{2+}$ have been argued as a key factor of the $I_{KACh}$ relaxation since this feature is abolished when the efficacious intracellular Ca$^{2+}$ chelator BAPTA is used [13]. However, we (here and in [12]) and other [31] have been able to reproduce the characteristic relaxation of $I_{KACh}$ despite the use of such compound. Interestingly, the mutant D$_{2s}$ S193A receptor, which practically annulled the voltage

dependence of $D_{2s}$ receptor with dopamine, also greatly decreases the relaxation, supporting the notion that this property is determined by the GPCR voltage sensitivity [31].

Altogether, our results provide additional support to the notion of the agonist-specific voltage dependence of GPCRs and that the voltage-dependent features of coupled effectors, such as the $I_{KACh}$ relaxation, are determined by this property.

## Supporting information

**S1 Table. Individual parameters obtained from the fits of the C-R relationships for Ado.** (DOCX)

**S2 Table. Individual parameters obtained from the fits of the C-R relationships for ACh.** (DOCX)

## Acknowledgments

The authors wish to thank Miguel Angel Flores-Virgen for technical assistance.

## Author Contributions

**Conceptualization:** José A. Sánchez-Chapula, Eloy G. Moreno-Galindo.

**Data curation:** Ana Laura López-Serrano, Rodrigo Zamora-Cárdenas, Iván A. Aréchiga-Figueroa, Pedro D. Salazar-Fajardo, Javier Alamilla, Eloy G. Moreno-Galindo.

**Formal analysis:** Ana Laura López-Serrano, Rodrigo Zamora-Cárdenas, Iván A. Aréchiga-Figueroa, Pedro D. Salazar-Fajardo, Tania Ferrer, Javier Alamilla, Eloy G. Moreno-Galindo.

**Funding acquisition:** Ricardo A. Navarro-Polanco, Eloy G. Moreno-Galindo.

**Supervision:** Tania Ferrer, Ricardo A. Navarro-Polanco, Eloy G. Moreno-Galindo.

**Writing – original draft:** Tania Ferrer, Ricardo A. Navarro-Polanco, Eloy G. Moreno-Galindo.

**Writing – review & editing:** Ana Laura López-Serrano, Rodrigo Zamora-Cárdenas, Iván A. Aréchiga-Figueroa, Pedro D. Salazar-Fajardo, Tania Ferrer, Javier Alamilla, José A. Sánchez-Chapula, Ricardo A. Navarro-Polanco, Eloy G. Moreno-Galindo.

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
