## [Decision Letter · Decision Letter 0]

9 Sep 2021

PONE-D-21-24751Differential voltage-dependent modulation of the ACh-gated K+ current by adenosine and acetylcholinePLOS ONE

Dear Marino-Galindo,

Thank you for submitting your manuscript to PLOS ONE. After careful consideration by three reviewers and myself, we feel that it has merit but does not fully meet PLOS ONE’s publication criteria as it currently stands. Therefore, we invite you to submit a revised version of the manuscript that addresses the all points raised during the review process.

In addressing the specific points of the reviewers, we ask that 1) a more rigorous statistical analysis be performed and described in the Methods and/or figure legends corresponding to the analyses, and 2) a tract be added to the Discussion which clearly and adequately distinguishes your current findings from earlier work published by your group.

We look forward to receiving your revised manuscript.

Kind regards,

Roger A. Bannister, PhD

Academic Editor

PLOS ONE

Journal Requirements:

[Doctoral Fellowships were awarded to A.L.L-S., R.Z-C., and P.D.S-F. (#286520, 587036, and 231965, respectively) from CONACYT, Mexico. The authors wish to thank Miguel Angel Flores-Virgen for technical assistance.]

 [This work was supported by SEP-CONACYT, Mexico. Grant No. CB-2011-01-167109 (to E.G.M-G.), and CB-2013-01-220742 (to R.A.N-P). The funders had no role in study design, data collection and analysis, decision to publish, or preparation of the manuscript.]

Reviewers' comments:

Reviewer's Responses to Questions

**Comments to the Author**

1. Is the manuscript technically sound, and do the data support the conclusions?

Reviewer #1: Yes

Reviewer #2: Yes

Reviewer #3: Yes

2. Has the statistical analysis been performed appropriately and rigorously? 

Reviewer #1: No

Reviewer #2: Yes

Reviewer #3: Yes

3. Have the authors made all data underlying the findings in their manuscript fully available?

Reviewer #1: Yes

Reviewer #2: Yes

Reviewer #3: Yes

4. Is the manuscript presented in an intelligible fashion and written in standard English?

Reviewer #1: Yes

Reviewer #2: Yes

Reviewer #3: Yes

5. Review Comments to the Author

Reviewer #1: In this paper, Lopez-Serrano examined the modulation of GIRK currents by Adenosine and ACh in guinea pig cardiac myocytes. The authors have shown in previous publications how ACh can modulate GIRK currents in a voltage-sensitive manner. In the current study, they compared Adenosine with ACh effects. Their results confirm the ACh effects, while Adenosine does not appear to modulate GIRK channels in a similar manner. Overall, the paper is well written and logically presented. However, the findings presented with ACh are not novel. The authors have shown similar findings previously, albeit in cat cardiac myocytes. They have not presented a clear justification of why they are replicating previous work. There is no insight provided (nor learned) on the mechanism by which adenosine may be different from ACh even though they use the same Gai/o G proteins.

In regards to Figure 1B, the authors state that the Adenosine C-R obtained at -100 mV was significantly different (P<0.03) than the C-R obtained at +30 mV. This does not seem to be the case and the authors do not state what statistical test they employed to arrive to this conclusion. They must provide the test to the reader, otherwise it is not convincing.

Reviewer #2: The manuscript "Differential voltage-dependent modulation of the Ash-gated K current by adenosine and acetylcholine," by Lopez-Serrano et al., describes a study aimed at determining whether membrane voltage alters adenosine receptor function. Previous work had shown the muscarinic receptors can exhibit altered function with strong changes in membrane potential, including changes in agonist affinity and/or binding. Here the authors examined adenosine receptors in cardiac myocytes to test whether they exhibited similar voltage dependent changes. They found that apparent agonist affinity (measured using I-KAch as an assay for receptor activity and current deactivation as a proxy for ligand-receptor unbinding) appeared unaltered. Authors also reported no change in I-KAch relaxation with membrane potential. By contrast, authors show that similar parameters were altered when currents were activated by acetylcholine, as previously reported, thus acting as a positive control. Ultimately the data provided were compelling that adenosine receptors do not show voltage dependence.

Minor issues:

In some figures (figs 2&5), authors display scatter plots illustrating raw data points, but in most figures they do not. Although it may clutter the figures, showing all of the raw data points would be preferable.

Fig. 4 shows dose-response curves for Ach responses. Curves do not seem to reach saturation, so it's unclear how accurate the EC50 fits are, nor whether changes in efficacy were evident (since data are only displayed as normalized to max).

Reviewer #3: This is an interesting report on how the activation of adenosine and acetylcholine receptors can modulate the current of K+ activated by acetylcholine (IKACh) in cardiac myocytes. In this work, the authors find a voltage-dependent differential modulation of IKACh by Ado and ACh. In the first case, the influence is null, while the second showed an apparent voltage-dependent effect on IKACh and induced a property known as "relaxation" of the current. Even though both agonists activate the same signaling pathway, the authors argue that both GPCRs show a different voltage sensitivity that modifies their affinity for the ligand that could help explain their differential effects on IKACh. In this reviewer's opinion, it is an interesting work, correctly planned and executed, clearly written in which the data support the authors' conclusions. However, I have a few considerations for the authors:

1) In the description of Figure 1B, the authors mention that the results of the analysis of the IKACh data obtained at -100 and +30 mV after applying Ado at different concentrations are statistically different. However, the data in the graph shows that this may not be the case. The data is very similar. This statement by the authors is actually contrary to their hypothesis (as shown by the results of Figs. 1-3) that Ado does not modify the properties of IKACh. For this reason, conducting a more rigorous and complete statistical analysis would strengthen the paper. This analysis should consider information on how the statistical analysis was conducted in Figs 1B and 4B (whether the t-test is the best option; and indicate the exact P values).

2) On the other hand, this reviewer would not recommend using scatter plots with means ± SD to represent experimental data (specifically in Figs. 2B and 5B). Instead, box-and-whisker plots that provide more information to summarize data should be used. The authors should replace these plots with box-and-whisker plots, with the box showing the median and the 25th and 75th quartiles and the whisker representing the 5th and 95th percentile. In addition, it would be nice if data points could be added to the plot superimposed on the box-and-whisker graphs. It sounds complicated, but fortunately, there are powerful statistical software packages available that can give these results directly.

3) Perhaps it would be worthwhile for the authors to add a paragraph to the discussion section regarding the characteristics, if known, in the molecular machinery that causes Ach receptors to have a higher affinity for the ligand at different voltages.

6. PLOS authors have the option to publish the peer review history of their article (what does this mean?). If published, this will include your full peer review and any attached files.

Reviewer #1: No

Reviewer #2: **Yes: **Paul J. Kammermeier

Reviewer #3: No

---

## [Decision Letter · Decision Letter 1]

25 Nov 2021

PONE-D-21-24751R1Differential voltage-dependent modulation of the ACh-gated K+ current by adenosine and acetylcholinePLOS ONE

Dear Dr. Moreno-Galindo,

Thank you for submitting your manuscript to PLOS ONE. After careful consideration, we feel that it has merit but does not fully meet PLOS ONE’s publication criteria as it currently stands. Therefore, we invite you to submit a revised version of the manuscript that addresses the points raised during the review process.  Given the marginal significance of the p value indicated for Ado in experiments shown in Fig. 1, we ask that the persistent points of Reviewer 1 be addressed as requested and/or that the implications of a borderline p value in this particular case be discussed. Please also tend to the minor change to Fig. 1B.

We look forward to receiving your revised manuscript.

Kind regards,

Roger A. Bannister, PhD

Academic Editor

PLOS ONE

Journal Requirements:

Reviewers' comments:

Reviewer's Responses to Questions

**Comments to the Author**

1. If the authors have adequately addressed your comments raised in a previous round of review and you feel that this manuscript is now acceptable for publication, you may indicate that here to bypass the “Comments to the Author” section, enter your conflict of interest statement in the “Confidential to Editor” section, and submit your "Accept" recommendation.

Reviewer #1: (No Response)

Reviewer #3: All comments have been addressed

2. Is the manuscript technically sound, and do the data support the conclusions?

Reviewer #1: Yes

Reviewer #3: Yes

3. Has the statistical analysis been performed appropriately and rigorously? 

Reviewer #1: No

Reviewer #3: Yes

4. Have the authors made all data underlying the findings in their manuscript fully available?

Reviewer #1: Yes

Reviewer #3: Yes

5. Is the manuscript presented in an intelligible fashion and written in standard English?

Reviewer #1: Yes

Reviewer #3: Yes

6. Review Comments to the Author

Reviewer #1: In the resubmission, the authors have clarified the reason for replicating previous studies.

Additionally, it was requested that the authors indicate the statistics used to conclude that the C-R curves in Figure 1B were "displaying a slight but statistically significant (P=0.03; paired t-test) greater potency at the negative potential". In the resubmission, the authors have provided 2 additional tables with the individual parameters generated by the fits for each cell. Why don't the authors run a test to determine whether the fit for one curve is different from the other curve? Also, why did the authors exclude the EC50 values in both Tables 1 and 2? Is there a reason why the authors chose to compare the difference in pEC50 for each cell instead of the EC50? Did they observe the same statistical differences when comparing EC50 values? Finally, the authors compared the differences of the pEC50 means, and those were significantly different. But the pEC50 values were not compared. So it is incorrect to state that there was a statistically significant greater potency.

Minor: For figure 1B, the authors should remove the individual raw data points and just show the mean values.

Reviewer #3: In the opinion of this reviewer, the authors have adequately addressed the comments raised in the previous round of review.

7. PLOS authors have the option to publish the peer review history of their article (what does this mean?). If published, this will include your full peer review and any attached files.

Reviewer #1: No

Reviewer #3: No

---

## [Editor Report · Decision Letter 2]

15 Dec 2021

Differential voltage-dependent modulation of the ACh-gated K+ current by adenosine and acetylcholine

PONE-D-21-24751R2

Dear Dr. Moreno-Galindo,

We’re pleased to inform you that your manuscript has been judged scientifically suitable for publication and will be formally accepted for publication once it meets all outstanding technical requirements.

Kind regards,

Roger A. Bannister, PhD

Academic Editor

PLOS ONE

Additional Editor Comments (optional):

Please revert to including the data points in Fig. 1, as originally requested by R2.
---

## [Editor Report · Acceptance letter]

21 Dec 2021

PONE-D-21-24751R2 

Differential voltage-dependent modulation of the ACh-gated K+ current by adenosine and acetylcholine 

Dear Dr. Moreno-Galindo:

I'm pleased to inform you that your manuscript has been deemed suitable for publication in PLOS ONE. Congratulations! Your manuscript is now with our production department. 

Kind regards, 

on behalf of

Dr. Roger A. Bannister 

Academic Editor

PLOS ONE